# Efficiency of Long Lateral Mass Screws

**DOI:** 10.3390/jcm11071953

**Published:** 2022-03-31

**Authors:** Seiya Watanabe, Kazuo Nakanishi, Kosuke Misaki, Kazuya Uchino, Hideaki Iba, Takachika Shimizu

**Affiliations:** 1Traumatology and Spine Surgery, Kawasaki Medical School, Okayama 701-0192, Japan; k.nakanishi@med.kawasaki-m.ac.jp (K.N.); k-misaki@med.kawasaki-m.ac.jp (K.M.); kazuya_u@med.kawasaki-m.ac.jp (K.U.); i-hideaki@med.kawasaki-m.ac.jp (H.I.); 2Gunma Spinal Hospital, Gunma 370-0871, Japan; tshimizu@dan.wind.ne.jp

**Keywords:** cervical spine, lateral mass screw, posterior fusion

## Abstract

Introduction: Lateral mass screws (LMS) have been widely used for the posterior fusion of the cervical spine. Even though LMS are safe, the screws are short and postoperative fixation is uncertain. Therefore, we measured and reported a technique using long lateral mass screws (LLMS), a new method of screw insertion, using a Zed spine from LEXI (Tokyo, Japan). Materials and Methods: In this study, we evaluated the outcomes of 35 patients who underwent surgery using LLMS at our hospital from 2019 to 2021. Operative time, blood loss, complications, inserted screw length, screw length based on gender differences, and screw deviation rate were evaluated. The Mann–Whitney U test was used to determine the gender differences in screw length. Screw deviation was evaluated by postoperative CT and a Zed spine to determine the screw insertion angle. Results: The mean operative time was 185 ± 51 min (120–327 min), and the mean blood loss was 236 ± 316 g (10–1720 g). The total number of screws was 183. The screw length was 22.2 (16–28) mm for males and 20.8 (16–28) mm for females, with an average length of 21 ± 2.7 mm. No gender differences were observed in terms of screw length (*p* > 0.01 NS). The number of deviated screws above G3 was one in the third cervical vertebra, three in the fourth cervical vertebra, one in the fifth cervical vertebra, and one in the sixth cervical vertebra. The number of deviated screws was 6 out of 183, and the deviation rate was 3.2%. Conclusions: In this study, the LLMS deviation rate was 3.2%, and strong fixation was possible without any complications. We measured the screw length and screw deviation rate in cases in which LLMS were actually inserted.

## 1. Introduction

Lateral mass screws (LMS) have been widely used for the posterior fusion of the cervical spine. LMS are safe, but the screws are short, and there are concerns about postoperative fixation. Therefore, we measured and reported a technique using long lateral mass screws (LLMS), a new method for inserting screws longer than LMS, using a Zed spine from LEXI (Tokyo, Japan). A Zed spine is a 3D preoperative planning software for posterior spinal fusion that uses CT images. The LLMS can be inserted without damaging the vertebral artery by inserting the screw at an angle of 50 degrees cephalad and 30 degrees outward [1] (Figure 1). This swing angle is difficult to achieve with conventional cervical screw swing angles. In addition, LLMS have the advantage of the insertion point and angle being predetermined to a certain extent, meaning that that insertion can be performed with fluoroscopy alone, without relying on navigation. When using fluoroscopy, the angle of insertion of the screw in the sagittal plane is parallel to the facet joint and is aimed at the angle between the posterior wall of the vertebral body and the superior border. We inserted the screws along the vertebral arch to avoid damaging the VA. However, one should be careful to not damage the cortical bone. This is the difficulty of inserting screws without navigation. Other than the screw length, the advantages of the LLMS are that the insertion point is medial to the LMS, so it does not need to be expanded to the outside, and that there is a graft matrix for lateral mass bone grafting. This means that the vertebral arch is not expanded laterally, thus reducing the amount of blood loss. The disadvantage of LLMS is that the screw heads are inserted much more medially than when using conventional lateral mass screws, making them difficult to use in conjunction with vertebroplasty. When performing a vertebral arch resection, it is necessary to drill the screw hole first and then perform decompression without inserting the screw.

## 2. Materials and Methods

We examined the outcomes of 35 patients who underwent surgery using LLMS at our hospital from 2019 to 2021. The average follow-up period was 17.9 ± 5.8 months (6–27 months). The mean age of the patients was 70 ± 11.3 years (32–88 years), and the sample included 22 males and 13 females. Reported diseases included 23 cases of cervical spinal cord injury, 5 cases of the ossification of the posterior longitudinal ligament of the cervical spine, 4 cases of cervical myelopathy, 1 case of destructive spondyloarthropathy, 1 case of a metastatic spinal tumor, and 1 case of rheumatoid arthritis. Angio-CT scan were not taken before the procedure because the screws are not interested into C1 and C2 level. The insertion point and path of the LLMS were measured from the third cervical vertebrae to the sixth cervical vertebrae using the Zed Spine in 50 patients who underwent surgery or CT for diagnostic purposes. The insertion point of the LLMS was 2 mm cranial and 2 mm medial to the medial margin of the intervertebral joint (Inter laminar V), and screws with a minimum length of 20 mm and a maximum length of 28 mm could be inserted. The difference in screw length from left to right was evaluated using the Wilcoxon test, but there was no significant difference (*p* > 0.01 NS). Furthermore, the LLMS were able to be inserted without damaging the vertebral artery by inserting the screws at an angle of 50 degrees cephalad and 30 degrees outward. Operative time, blood loss, complications, screw length, and screw deviation rate were evaluated. The Mann–Whitney U test was used to determine any gender-related differences in the screw length. To evaluate screw deviation, we examined the screw insertion angle by means of postoperative CT and the Zed spine. For screw deviation, we classified the screws that could be inserted without any problems as G0, and those that deviated as G1 to G4, with G1 being a slight perforation of the vertebral artery wall, G2 being a perforation that was less than 1/2 of the screw diameter, G3 being a perforation of more than 1/2 of the screw diameter, and G4 as a perforation that was the same size as entire screw diameter (Figure 2). Categories G3 and above were defined as deviations.

## 3. Results

The results showed that the mean operative time was 185 ± 51 min (120–327 min), and the mean blood loss was 236 ± 316 g (10–1720 g). There were 183 screws. The screw length was 22.2 mm (16–28) for males and 20.8 mm (16–28) for females, with an average length of 21 ± 2.7 mm. No gender difference in screw length was observed (*p* > 0.01 NS). The number of deviated screws above G3 was one in the third cervical vertebra, three in the fourth cervical vertebra, one in the fifth cervical vertebra, and one in the sixth cervical vertebra. A total of 6 out of 183 screws deviated. The screw deviation rate was 3.2%. No complications, such as neurovascular injuries, (Table 1) occurred. Severe deviations occurred in three cases. However, these cases were discharged from the hospital without foot stroke. However, we noted one case of postoperative wound infection. The average angle of insertion to the outside of the Grade 0 screw was 29°, which was close to the ideal insertion angle of 30° examined in the Zed spine. However, the deviated screws had a smaller outward insertion angle. The G3 and G4 screws had an insertion angle of 18° (Table 2).

(Case)

A 69-year-old female patient had obtained a central cervical spinal cord injury due to a fall. A simple cervical CT scan showed the segmental ossification of the cervical posterior longitudinal ligament from the second to the sixth cervical vertebrae, and an MRI scan showed narrowing of the spinal canal and spinal cord compression in the same region (Figure 3). LLMS were inserted in the third through sixth cervical vertebrae. A 3.5 × 28 mm screw was inserted in the third and fourth cervical vertebrae, and a 3.5 × 26 mm screw was inserted in the fifth and sixth cervical vertebrae. The operative time was 233 min, and the blood loss was 165 g. A postoperative simple CT scan of the cervical spine showed good screw placement (Figure 4).

## 4. Discussion

The purpose of internal fixation is to stabilize the spine and correct deformities. In the literature, the first cervical internal fixation was reported in 1891 by Hadra [2,3]. Hadra used silver wire to repair dislocated fractures. In 1937, Gallie [4] reported the wire fixation of the cervical spine. Rogers [5] and Murphy [6] reported interspinous process wiring and several similar methods. Additionally, in the late 1980s, Roy–Camille [7] reported a method using screws inserted into the lateral mass of the cervical spine as anchors, which provided better fixation than wire fixation and could be used in combination with laminectomy. The lateral mass screw (LMS) method has since become a safe method and one of the main techniques for the posterior fusion of the cervical spine, as reported by Magerl [8] and Anderson [9]. Furthermore, the use of LMS technique has been reported to be able allow insertion into the atlas [10]. Cervical pedicle screws (PS) were first reported by Leconte [11] in 1964. In Japan, Abumi et al. [12] advocated for the safety of PS and reported that their fixation strength was superior to other cervical posterior fusions [13,14] and that they could be used for various conditions. Refs. [15,16] shows that PS have a high pull-out strength because the screws are inserted into the vertebral arch root, which has a thick cortical bone. The pullout strengths were significantly higher for the cervical pedicle screws than they were for the lateral mass screws (1214 vs. 332 N) [14]. PS fixation was determined to be very strong. However, if the screw deviates from the pedicle, major complications such as nerve damage and vertebral artery damage may occur. In addition, PS cannot be inserted in cases with narrow pedicles or unilateral vertebral artery injury. The advantage of LMS is the low incidence of neurovascular injury due to screw deviation. Nerve root injury attributed to screw placement only occurs in 1.0% of cases, and no cases of vertebral artery injury have been reported. Ref. [17] shows that in cases where iatrogenic severe vertebral artery injuries occur, the symptomatic stroke and mortality rates have been reported to be 0% to 50% and 0% to 16%, respectively [18]. However, there are concerns about fixation strength in patients who are older and with osteoporosis, and it has been reported that intraoperative fracture of the lateral mass screws and postoperative dislocation may occur [19]. Another institution reported that postoperative screw dislocation occurred in 3 of 143 cases (2.1%) [20]. In this study, no operative screw dislocation occurred. In recent years, the evolution of implants has widened the angle of screw insertion. In the past, the swing angle of the cervical screw head was about 30 to 40 degrees. New screw heads have a swing angle of more than 50 degrees, which enables fixation, even when the screw is inserted at a considerable angle. Therefore, we presented a method for inserting long lateral mass screws (LLMS), which are stronger than LMS and safer than PS, even in patients with severe osteoporosis [1]. The LMS length was 14–15 mm using the Roy–Camille method [7] and 15–16 mm in the Magerl method [8]. In the Zed spine, the mean screw length of the LLMS was 24 mm. However, the average inserted screw length was 21 mm. The deviation rate for pedicle screws has been reported to range from 1.2% to 18.7% when using navigation [21,22]. In this study, the deviation rate of the LLMS was 3.2%, and strong fixation was possible without any complications. The deviation angle in the G3 and G4 screws was 18°. The cause of these deviations may have been due to insufficient resection during the spinous process. Insufficient resection during the spinous process prevents the screw from tilting outward. Another possible cause is that the screws are unable to be sufficiently titled in cases where there is thick soft tissue in the back. We measured the screw length and screw deviation rate in cases in which the LLMS were actually inserted. We did not evaluate BMD. In the future, it will be necessary to evaluate the BMD and consider the pull-out strength of the screws. Furthermore, the limitations of this study are that we did not analyze the number of cases and did not compare LLMS to the LMS control group.

## 5. Conclusions

We reviewed cases of LLMS. The mean screw length of the LLMS was 21 mm. The screw deviation rate of the LLMS was 3.2%. We were not able to achieve a swing angle of 30 degrees in cases where there was severe deviation. The ideal LLMS insertion angle was fine for the Zed spine measurements.

## Figures and Tables

**Figure 1 jcm-11-01953-f001:**
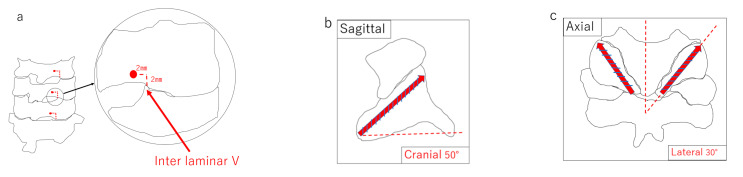
(**a**) Insertion point. Lateral margin between the vertebral arches (Inter laminar “V”). (**b**) Sagittal cross-sectional image of screw insertion using the screw as a reference. (**c**) Axial cross-sectional image of screw insertion using the screw as reference.

**Figure 2 jcm-11-01953-f002:**
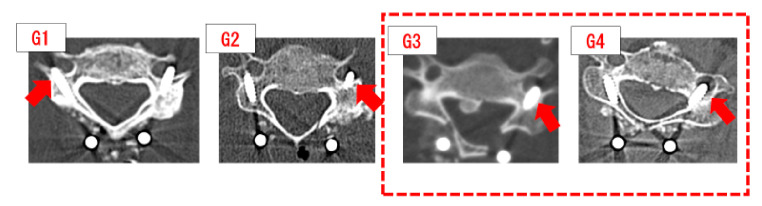
Degrees of screw deviations: (**G1**) Screw only slightly breaks through the inner side of the vertebral artery; (**G2**) penetration of less than 1/2 of the screw diameter; (**G3**) more than 1/2 of the screw diameter has breached the vertebral artery; (**G4**) the entire screw diameter has breached the vertebral artery.

**Figure 3 jcm-11-01953-f003:**
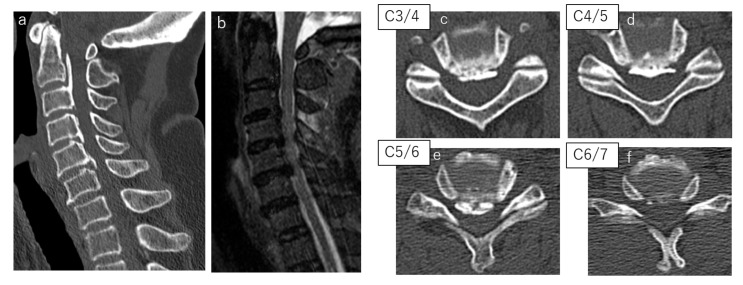
(**a**) preoperative CT sagittal image; (**b**) preoperative MRI image (T2); (**c**) preoperative CT axial image (C3/4); (**d**) preoperative CT axial image (C4/5); (**e**) preoperative CT axial image (C5/6); (**f**) preoperative CT axial image (C6/7).

**Figure 4 jcm-11-01953-f004:**
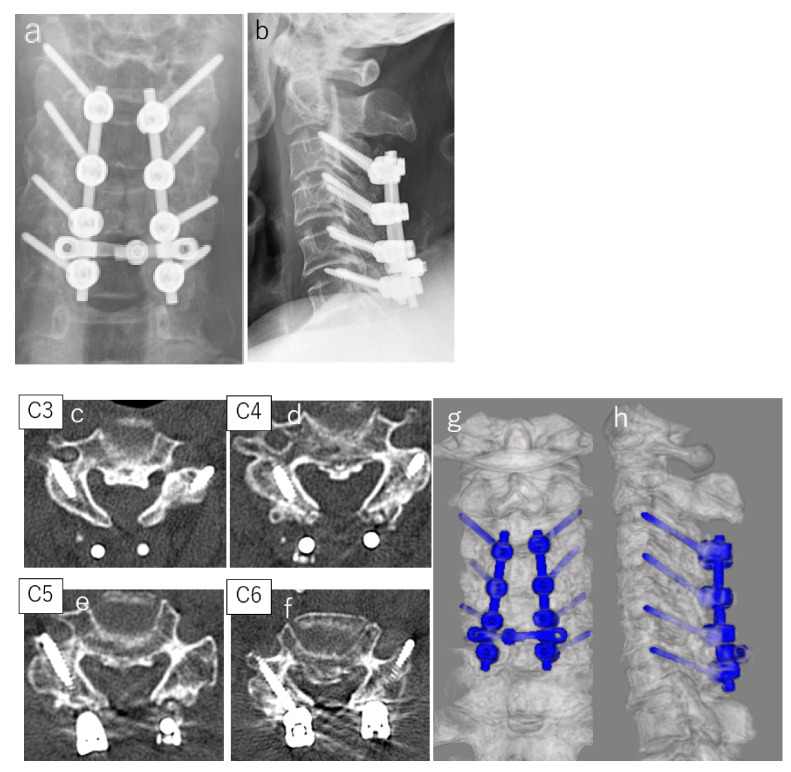
(**a**) postoperative X-ray frontal image; (**b**) lateral postoperative X-ray image; (**c**) postoperative CT axial image (**C3**); (**d**) postoperative CT axial image (**C4**); (**e**) postoperative CT axial image (**C5**); (**f**) postoperative CT axial image (**C6**); (**g**) postoperative 3DCT frontal image; (**h**) postoperative 3DCT lateral image.

**Table 1 jcm-11-01953-t001:** Screw deviation rate.

6/183 Screws (3.2%)		
	Right	Left
C3	1	0
C4	1	2
C5	1	0
C6	0	1

Deviation was defined as G3 or G4.

**Table 2 jcm-11-01953-t002:** Screw insertion angle.

	Angle
Grade0 (143)	29°
Grade1 (20)	22°
Grade2 (14)	19°
Grade3 (3)	18°
Grade4 (3)	18°

## Data Availability

Not applicable.

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
