# Peer review of "Efficiency of Long Lateral Mass Screws"

_jcm, 2022, doi:10.3390/jcm11071953_

Round 1
Reviewer 1 Report
ABSTRACT
- English should be reviewed
- Blood loss and operative time data should be compared to similar cases using “normal” length lateral mass screws
INTRODUCTION
- English should be reviewed
- Parts of the introduction seem more suitable to be put in the material and methods part (lines 31-39)
- The authors should explain what a ZED spine is, not everybody may understand
- What is the difference between the long lateral mass screws and the “normal” ones should be better explained
MATERIALS and METHODS
- English should be reviewed (… the screw length of gender difference… ; this sentence, for example, isn’t well structured)
- Lines 71-77: the classification in G0-G4 concerning the deviation of the screw isn’t clear and is repeated
RESULTS
- English should be reviewed
- The results should be compared to a control group with normal lateral mass screws
- The authors state they had ZERO complications, is that possible?
(Representative case)
- Line 106: LLMS WERE inserted…
DISCUSSION
- English should be reviewed
- Lines 123-127: repetition “the aim of internal fixation is stability and deformity correction”.. the first two sentences can be unified in 1 sentence
- Lines 131-133: repetition
- Lines 152-154: repetition
Reviewer 2 Report
Watanabe et al. present a primary manuscript regarding the efficience of long lateral mass screws for cervical spine surgery.
Abstract: "In a case where the screw was actually inserted" - this sentence does not make sense at this point.
Introduction: The Zed spine needs more detailed explanation at the first appearance.
line 41-44: This is controversial; some may suggest that every angle is determined "to some extent". And If you choose to state that it is not needed to "rely on navigation", this should be referenced explicitly and elaborately.
Fig. 1 should be redrawn, not just used from another reference. Additionally, the copy-pase of the reference is unacceptable.
Materials and Methods
How was the needed sample size calculated, did you perform a power analysis a priori?
The inclusion of metastatic spinal tumor reduced the overall power since the study population is very heterogeneous. What is your rationale for that?
I am missing the ASA status in the patient cohorts.
Why was Mann-Whitney U test used, did you check for normal distribution before?
Please clarify whether an Angio-CT has been done before surgery in order to exclude a high-riding vertebral artery.
Results
Table 1: Is "deviation" referring to at least G1 or G3? Please clarify in the table legend.
Table 2: 3 were G3 and 3 G4? So these seem like substantial complications. Did you thoroughly examine the patients status postoperatively?
Fig. 3: Please show a T1 sagittal image as well (might be Fig. 3c).
Fig. 4b: The sagittal image is unsatisfying, since not exactly sagittal.
Fig. 4e: Please grade the rate of deviation in C5.
Fig. 4f: How do you justify the screw length in left C6?
Discussion
The discussion needs to be thoroughly redesigned and rewritten. Right now, it is merely more than a historical overview paired with highlighting head swing screws.
The first historical parts are of interest, but do not add much to the overall message. I would suggest to get rid or at least shorten the first parts substantially.
Please focus on a deeper literature review and comparison.
Please report more about the possible (and devastating) effects of occuring arterial injury. Please report general numbers and how rigorously this has been reported and reviewed in the literature.
Likewise, a more detailed overview about the biomechanical literature regarding cervical pedicle screws needs to be added and reported about.
Osteoporosis: Did you screen your patients for osteoporosis or do you have the BMD of your patients?
In addition, did you assess complications? How and how long were the patients monitored?
Round 2
Reviewer 1 Report
ABSTRACT
- English is correctly reviewed
INTRODUCTION
- English has improved but it still needs a professional revision
MATERIALS and METHODS
- English isn’t yet of appropriate level
- The classification in G0-G4 concerning the deviation of the screw is now clear
RESULTS
- English should be reviewed
- No control group with normal lateral mass screws was described
DISCUSSION
- English can be improved
- Repetitions were removed
Reviewer 2 Report
The information about Zed spine is still insufficient.
Please add an information on the difficulty of inserting screws without navigation and how this is seen in the literature.
Also, having Tab. 2, your method does not seem to be without error.
Thank you for correcting fig. 1.
"We considered the number of cases as large as possible." Then please add to the limitations that you did not do a power analysis before, and give a proper reason for that.
The n=3 G3 and G4 need further evaluation. If so, please state at least that you performed CT analyses and a thorough clinical examination making sure that your misplacement did not affect postoperative status.
Fig. 3 needs an ap and a sagittal image (maybe an additional coronal view), not just one.
Fig. 4b: The exact sagittal image is fine now.
Fig. 4e: Please comment on the potential vascular affection of the right C5 screw.
Fig. 4f: Since this is not an optimal image, I would suggest to replace this one.
Discussion:
How were the patients selected who "might need treatment"?
The lack of BMD assessment is a limitation and should be reported as one.
How long was the postoperative follow-up? This should be exactly reported (with SD or SEM) and the range.
The conclusions should be complete sentences, not bullet points.
I am still missing a deep discussion on lateral mass screws, and a delineation of your study to the subaxial spine.
